# Mucosal absorption of therapeutic peptides by harnessing the endogenous sorting of glycosphingolipids

Maria Daniela Garcia-Castillo[1†], Daniel J-F Chinnapen[1,2,3†],
Yvonne M te Welscher[1], Rodrigo J Gonzalez[4,5], Samir Softic[6], Michele Pacheco[1],
Randall J Mrsny[7], C Ronald Kahn[6], Ulrich H von Andrian[4,5], Jesper Lau[8],
Bradley L Pentelute[9], Wayne I Lencer[1,3,2]*

[1]Division of Gastroenterology, Boston Children's Hospital, Boston, United States; [2]Department of Pediatrics, Harvard Medical School, Boston, United States; [3]Harvard Digestive Diseases Center, Boston, United States; [4]Department of Microbiology and Immunobiology, Harvard Medical School, Boston, United States; [5]Center for Immune Imaging, Harvard Medical School, Boston, United States; [6]Joslin Diabetes Center and Harvard Medical School, Boston, United States; [7]Department of Pharmacy and Pharmacology, Univeristy of Bath, Bath, United Kingdom; [8]Novo Nordisk, Måløv, Denmark; [9]Department of Chemistry, Massachusetts Institute of Technology, Cambridge, United States

*For correspondence:
Wayne.Lencer@childrens.harvard.edu

[†]These authors contributed equally to this work

**Abstract** Transport of biologically active molecules across tight epithelial barriers is a major challenge preventing therapeutic peptides from oral drug delivery. Here, we identify a set of synthetic glycosphingolipids that harness the endogenous process of intracellular lipid-sorting to enable mucosal absorption of the incretin hormone GLP-1. Peptide cargoes covalently fused to glycosphingolipids with ceramide domains containing C6:0 or smaller fatty acids were transported with 20-100-fold greater efficiency across epithelial barriers in vitro and in vivo. This was explained by structure-function of the ceramide domain in intracellular sorting and by the affinity of the glycosphingolipid species for insertion into and retention in cell membranes. In mice, GLP-1 fused to short-chain glycosphingolipids was rapidly and systemically absorbed after gastric gavage to affect glucose tolerance with serum bioavailability comparable to intraperitoneal injection of GLP-1 alone. This is unprecedented for mucosal absorption of therapeutic peptides, and defines a technology with many other clinical applications.

DOI: https://doi.org/10.7554/eLife.34469.001

## Introduction

One of the major challenges for applying protein and peptide biologics to clinical medicine is the lack of rational and efficient methods to circumvent epithelial and endothelial cell barriers separating large molecules from target tissues. In the case of epithelial cells lining mucosal surfaces, the pathway for absorption of large solutes is by transcytosis – a process of transcellular endosome trafficking that connects one surface of the cell with the other (*Tuma and Hubbard, 2003*; *Mostov et al., 2000*; *Garcia-Castillo et al., 2017*). The same is true for transport of protein and peptide cargoes across tight endothelial barriers that separate blood from tissue - typified by the blood-brain barrier (*Abbott, 2013*; *Pardridge, 2015*; *Preston et al., 2014*; *Lajoie and Shusta, 2015*). Here, we address these problems by testing structure-function of the glycosphingolipids for their intracellular

**eLife digest** To work properly, drugs need to be absorbed efficiently into the body. Medications that are injected directly into the bloodstream are often quickly transported to the organs or tissues they target. But injections are not always convenient, and many patients would instead prefer to swallow a pill or tablet. If a drug is swallowed, however, it must first be absorbed through the gut before it can enter the bloodstream.

The lining of the gut consists of tightly linked layers of cells that readily take up small molecules, such as water and simple nutrients, but exclude almost all larger ones. Since several important types of drugs are large or poorly absorbed molecules, such as proteins, finding methods to help them cross the gut barrier is a major part of drug development.

Originally from bacteria, cholera toxin is an example of a large, naturally occurring protein that does cross the gut lining. To do this, the toxin specifically attaches onto GM1, a type of lipid molecule that is found on the outer surface of gut cells, and hijacks the system that moves this lipid within cells. Previous studies identified several key features of GM1's structure that enable this movement; and, in 2014, researchers tested GM1 as a 'carrier' to help the gut to absorb large therapeutic molecules. This approach was successful in cells grown in the laboratory, but not when the drugs were fed to animals.

To overcome this issue, Garcia-Castillo, Chinnapen et al. – who include some of the researchers involved in the earlier studies – set out to further boost GM1's ability to transport drugs across the gut lining. First several hybrid molecules were made, consisting of different structures of GM1 (the 'carrier') fused to a reporter peptide (the 'cargo'). Laboratory experiments with human intestinal cells and dog kidney cells, both of which form tightly-linked layers much like the actual lining of the gut, revealed specific structural variations of the GM1-derived carrier that transported the cargo across the cell barrier more efficiently.

Garcia-Castillo, Chinnapen et al. went on to test the efficiency of these carriers further by switching the reporter cargo to a therapeutic hormone called GLP-1. This hormone is used to treat people with type II diabetes but is currently given via an injection. The same structural variants of GM1 that enhanced delivery of the reporter cargo also worked for the larger GLP-1 hormone. Garcia-Castillo, Chinnapen et al. then fed the GM1-GLP-1 fusions to mice, and measured the amount of GLP-1 hormone absorbed into the blood. Crucially, the mice fed GM1-GLP-1 molecules absorbed the drug just as well as mice injected with the GLP-1 that is normally given to diabetes patients.

Together these findings represent a major contribution to the pharmaceutical toolbox. They may also ultimately lead to more drugs that can be given as a patient-friendly pill or tablet, readily cross the gut barrier and achieve widespread drug delivery around the body.
DOI: https://doi.org/10.7554/eLife.34469.002

trafficking in transcytosis, and for their use as vehicles to enable transcellular transport of therapeutic peptides.

These studies were informed by our findings that the structure of the ceramide (lipid) domain plays a decisive role in the intracellular trafficking of the glycosphingolipid GM1, the lipid receptor responsible for cholera toxin entry into the endoplasmic reticulum (ER) of host cells and required for disease (*Chinnapen et al., 2012*). GM1 species containing ceramides with 'kinked' *cis*-unsaturated C18:1 or C16:1 fatty acids, or non-native 'short chain' C12:0 fatty acids, enter the sorting/recycling endosome of epithelial cells allowing for transport to various intracellular destinations: including the recycling pathway and retrograde pathway to the Golgi and ER. These lipids do not efficiently traffic into the late endosome-lysosome pathway. In contrast, GM1 sphingolipids with long saturated fatty acid chains (C16:0 or longer) sort almost exclusively into late endosomes and lysosomes (*Chinnapen et al., 2012*). The sorting step separating the intracellular distributions of these closely related lipids emerges from the early sorting endosome, and we find it robust across all cell lines so far tested. Our observations are consistent with the two major models for lipid sorting: one by molecular shape (*Hao et al., 2004*; *Mayor et al., 1993*; *Mukherjee et al., 1999*) and the other by

membrane microdomains (lipid rafts) (*Brown, 2006*; *Simons and Ehehalt, 2002*; *Simons and Vaz, 2004*).

In polarized epithelial cells, another pathway emerges from the sorting endosome and leads to membrane transport across the cell by transcytosis. The same GM1 species with *cis*-unsaturated or short-chain fatty acids that efficiently enter the recycling endosome also sort into this pathway (*Saslowsky et al., 2013*; *te Welscher et al., 2014*). By analogy with the bacterial toxins and viruses that bind glycosphingolipids for trafficking into host cells (*Chinnapen et al., 2007*; *Spooner and Lord, 2012*; *Ewers and Helenius, 2011*; *Cho et al., 2012*), this result suggested a means for enabling the uptake and transepithelial transport of protein or peptide therapeutic cargoes for mucosal delivery. Our first attempt to test this idea showed that these glycosphingolipid species were capable of sorting a therapeutic cargo into the transcytotic pathway. But release into solution to effect transport across epithelial barriers in vitro, or absorption into the systemic circulation in vivo was not detectable (*te Welscher et al., 2014*). To solve this problem, we conducted additional structure-function studies for the glycosphingolipids in intracellular sorting and discovered modifications of the ceramide and oligosaccharide domains that enable the lipids to act as molecular carriers for mucosal absorption of therapeutic peptides, achieving levels of bioavailability comparable to that of intraperitoneal injection.

## Results

### Structure-function studies on the ceramide domain of GM1

To test if GM1 glycosphingolipids can be harnessed for biologic drug delivery, we first developed a non-degradable all D-isomer reporter peptide for structure-function studies on the ceramide domain. The reporter peptide was designed to contain two functional groups, a biotin for high-affinity streptavidin-enrichment, and an alkyne reactive group for chemical ligation to fluorophore molecules and quantitative detection. A C-terminal reactive aminooxy was used for coupling the reporter peptide to the oligosaccharide domain of the different GM1 species (*Figure 1A* and *Figure 1—figure supplement 1A*) (*te Welscher et al., 2014*). The functional groups on the reporter peptide, that is biotin, alkyne, fluorophore and combinations of, were tested to verify the absence of confounding effects on GM1 trafficking (*Figure 1—figure supplement 1B*). This was assessed by confocal microscopy for endosome sorting and transcytosis, using fluorescent cholera toxin B-subunit to label the GM1-peptide fusion molecules (*Figure 1—figure supplement 1C*). In all cases, the peptide-coupled GM1 species containing *cis*-unsaturated or short fatty acid ceramide domains sorted into small cytoplasmic vesicles and basolateral membranes consistent with the recycling and transcytotic pathways, whereas the peptide-coupled GM1 species containing saturated long fatty acid ceramide domains did not; they were sorted into larger cytoplasmic puncta consistent with the late endosome/lysosome instead (*Figure 1—figure supplement 1C*). Both events were blocked at 4°C consistent with uptake by endocytosis. These results are consistent with our previous studies (*Chinnapen et al., 2012*; *te Welscher et al., 2014*) and validate the reporter construct.

All glycosphingolipid-peptide fusion molecules subsequently prepared were coupled to Alexa Fluor-488 (AF488), purified by HPLC, and structures confirmed by mass spectrometry (*Figure 1—figure supplement 1A* and Material and methods). When tested by pulse-chase in MDCK cells, the peptide-GM1 fusion molecules were internalized and sorted as predicted (*Chinnapen et al., 2012*). The GM1 species containing long saturated fatty acids (C16:0-GM1) were localized to intracellular puncta consistent with sorting to the lysosome (*Figure 1B*, bottom panels), and the GM1 species containing short fatty acids were sorted into the recycling and transcytotic pathways as evidenced by localization to apical and basolateral plasma membranes and small intracellular vesicles (*Figure 1B* middle panels). This interpretation was confirmed using lysotracker to mark the lysosome and the transferrin receptor to mark the recycling endosome (*Figure 1—figure supplement 1D*). The peptide alone did not bind or enter cell monolayers (*Figure 1B*, top panels). Because we use GM1 originally purified from bovine brain to synthesize the different GM1 species, the end products comprise two isoforms of the long chain base: one containing a sphingosine chain of C18:1 and the other of C20:1. For the GM1 species containing C12:0 fatty acids, the two sphingosine-isoforms were purified and found to track identically in transcytosis (*Figure 1B*, middle two panels). Thus, it is the structure of the fatty acid that dominates in the sorting reactions (*Chinnapen et al., 2012*).

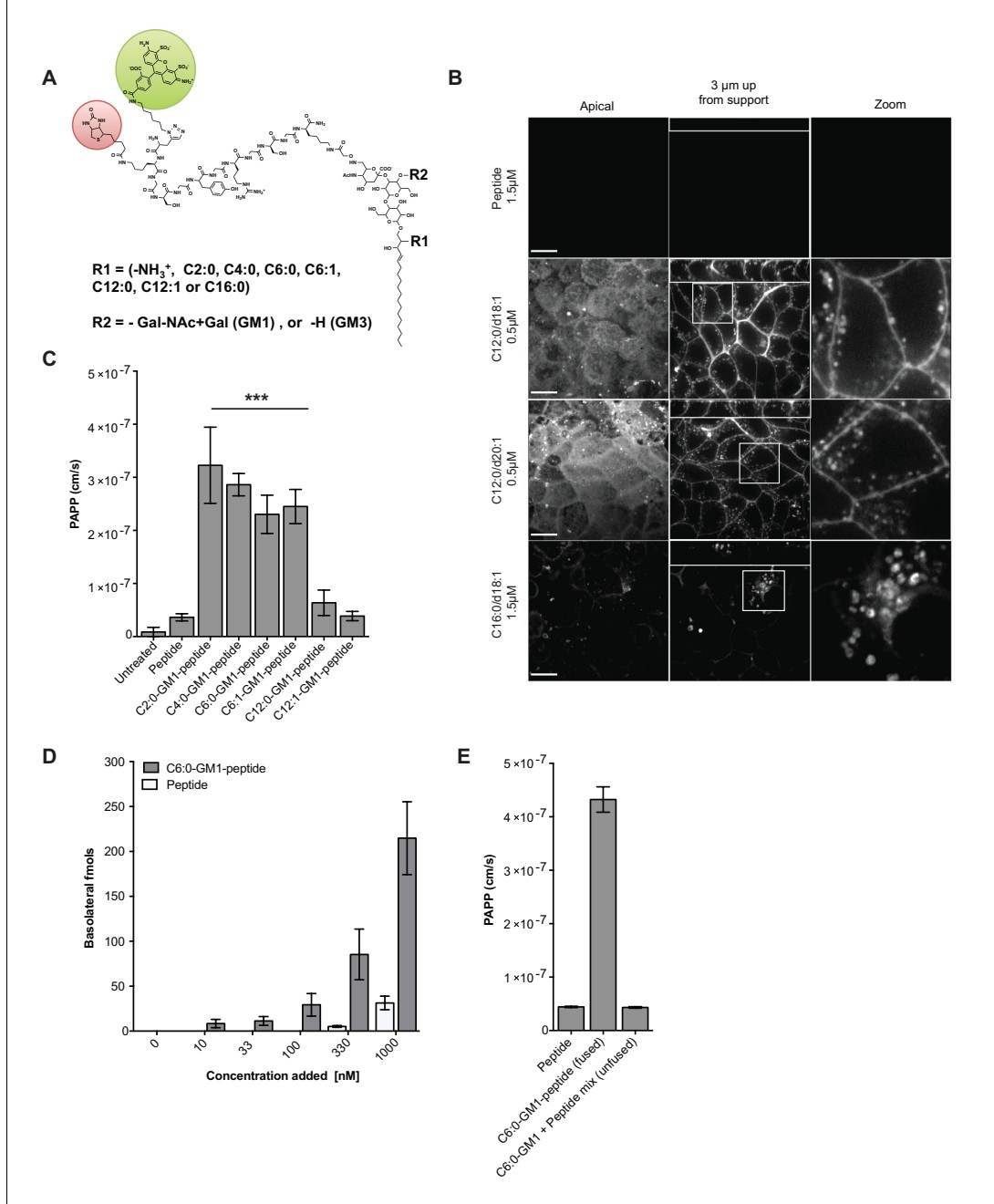

**Figure 1.** Modifications to the ceramide domain of GM1 results in enhanced trans-epithelial transport. (**A**) Representative structure of GM1 sphingolipids fused to an all-D amino acid reporter peptide. The reporter peptide contains a lysine-linked biotin (red circle) used for affinity purification and an N-terminal Alexa Fluor 488 (green circle). (**B**) GM1-peptide fusions or unfused reporter peptide were added apically to MDCK-II cells grown on filter supports and imaged by live cell confocal microscopy. Transcytosis of the C12:0-GM1-reporter peptide fusion (with either a C18:1 or C20:1 sphingosine) is evident by basolateral membrane fluorescence. The C16:0-GM1 reporter molecule is delivered to intracellular puncta, presumably lysosomes. Scale bars 10 um (**C**) Transport of the indicated GM1-reporter peptide fusions across T84 cell monolayers with the indicated fatty acid chain length and degree of saturation. The GM1-peptide fusions containing ceramide domains with short fatty acids have a ~10 fold increase in trans-epithelial transport over the unfused reporter peptide. (**D**) Transepithelial transport of the C6:0-GM1 peptide fusion across T84 monolayers is dose-dependent and far exceeds transport of the unconjugated reporter peptide. (**E**) T84 monolayers were simultaneously treated with unfused peptide and unfused C6:0-GM1. Mixing experiments confirm that fusion of the reporter peptide to the glycosphingolipid carrier is required for amplified transcellular transport.

DOI: https://doi.org/10.7554/eLife.34469.003

The following figure supplements are available for figure 1:

*Figure 1 continued on next page*

*Figure 1 continued*

**Figure supplement 1.** Synthesis and modifications to the reporter peptide and ceramide domain of GM1.
DOI: https://doi.org/10.7554/eLife.34469.004
**Figure supplement 2.** Validation of the transcytosis assay.
DOI: https://doi.org/10.7554/eLife.34469.005

To test structure-function of the ceramide fatty acid chain, we developed a quantitative assay for transcytosis (*Figure 1—figure supplement 2A–B* and Material and methods). The assay is sensitive to picomolar concentrations and linear over a large 6-log dynamic range (*Figure 1—figure supplement 2B*). Different GM1-peptide fusions (0.1 µM) were applied to apical reservoirs of polarized epithelial cell monolayers and transport to basolateral reservoirs analyzed after 3 hr by streptavidin-capture and quantitative fluorometric read (*Figure 1C*, *Figure 1—figure supplement 2C*). Defatted bovine serum albumin (1% w/v) was added to the basolateral reservoir to amplify release of the lipid-peptide fusion molecules from membrane to solution after transcytosis. In all studies, conditions for equal loading of the different GM1-peptide fusion molecules were determined by quantitative fluorescence measurement of washed cells treated with trypsin to remove adherent glycosphingolipids not incorporated into the membrane bilayer (*Figure 1—figure supplement 2D*).

Transcytosis for the different GM1-peptide fusion molecules was quantified as an apparent permeability coefficient (PAPP; cm/s) and compared against both the unfused reporter peptide (labeled peptide) or untreated monolayers as negative controls (*Figure 1C*, *Figure 1—figure supplement 2C*). When tested on human intestinal T84 cell monolayers, we found approximately a 10-fold increase in transepithelial transport (PAPP) for the GM1 ceramide species containing C6:0, C4:0, C2:0, fatty acids or lyso-GM1 as compared to controls. Introduction of an unsaturated *cis*-double bond to the short chain ceramide fatty acids (C12:1 and C6:1) had no effect on transcytosis in comparison to the saturated species (C12:0 and C6:0) (*Figure 1C*). This result is in contrast to the dramatic effect the *cis*-double bond induces in trafficking of the long fatty-acid chain GM1 glycosphingolipids (*Chinnapen et al., 2012*; *Saslowsky et al., 2013*). Transepithelial transport was dose-dependent for the C6:0-GM1-peptide fusion (grey bars) and greatly exceeded transport of the unconjugated reporter peptide (white bars) over a wide range of concentrations (*Figure 1D*). Mixing experiments using unconjugated GM1 and reporter peptide as individual molecules confirmed that transcellular transport of the peptide cargo was dependent on fusion to the GM1 glycosphingolipid (*Figure 1E*). Neither the unfused reporter peptide nor the GM1-peptide fusion had any detectable confounding effects on cell viability as determined by measurement of metabolic activity (MTT assay), or monolayer integrity and tight junction function assessed as trans-epithelial resistance (TEER) or dextran flux (*Figure 1—figure supplement 2E–G*).

## Active transport of the GM1-peptide fusions by transcytosis

Several approaches were used to confirm that the mechanism of cargo transport across epithelial cell monolayers was by transcytosis and not by paracellular leak. First, we tested for transport across epithelial monolayers at 4°C. Such low temperature effectively stops all forms of membrane dynamics including transcytosis, but has minimal effects on paracellular solute diffusion. We found no detectable transport of GM1-peptide fusions across T84 cell monolayers at 4°C, consistent with transport via transcytosis (*Figure 2A*). The same results were obtained when transcytosis was measured by live cell confocal microscopy. In these experiments, the apical membranes of epithelial cell monolayers were incubated with the C6:0-GM1-peptide fusion at 10°C for 45 min to allow for incorporation of the GM1 ceramide into the apical membrane with minimal uptake into the cell by endocytosis (*Figure 2B*, x–z and y–z images). Monolayers were washed and then chased for 15 min at 37°C or kept at a restrictive temperature of 10°C. We found the C6:0-GM1-peptide fusion localized to basolateral membranes in cells chased at 37°C, but not at 10°C (*Figure 2B*; left and middle panels respectively). Only after breaking open tight junctions by removal of extracellular Ca$^{2+}$ (EDTA treatment) did the GM1-peptide fusion molecule gain access to the basolateral membrane at 10°C (*Figure 2B*, right panel).

In a third approach, we blocked endocytosis at physiologic temperature using the dynamin inhibitor Dyngo-4A. For the C6:0 and C12:0 GM1-peptide fusion molecules, transport into the basolateral

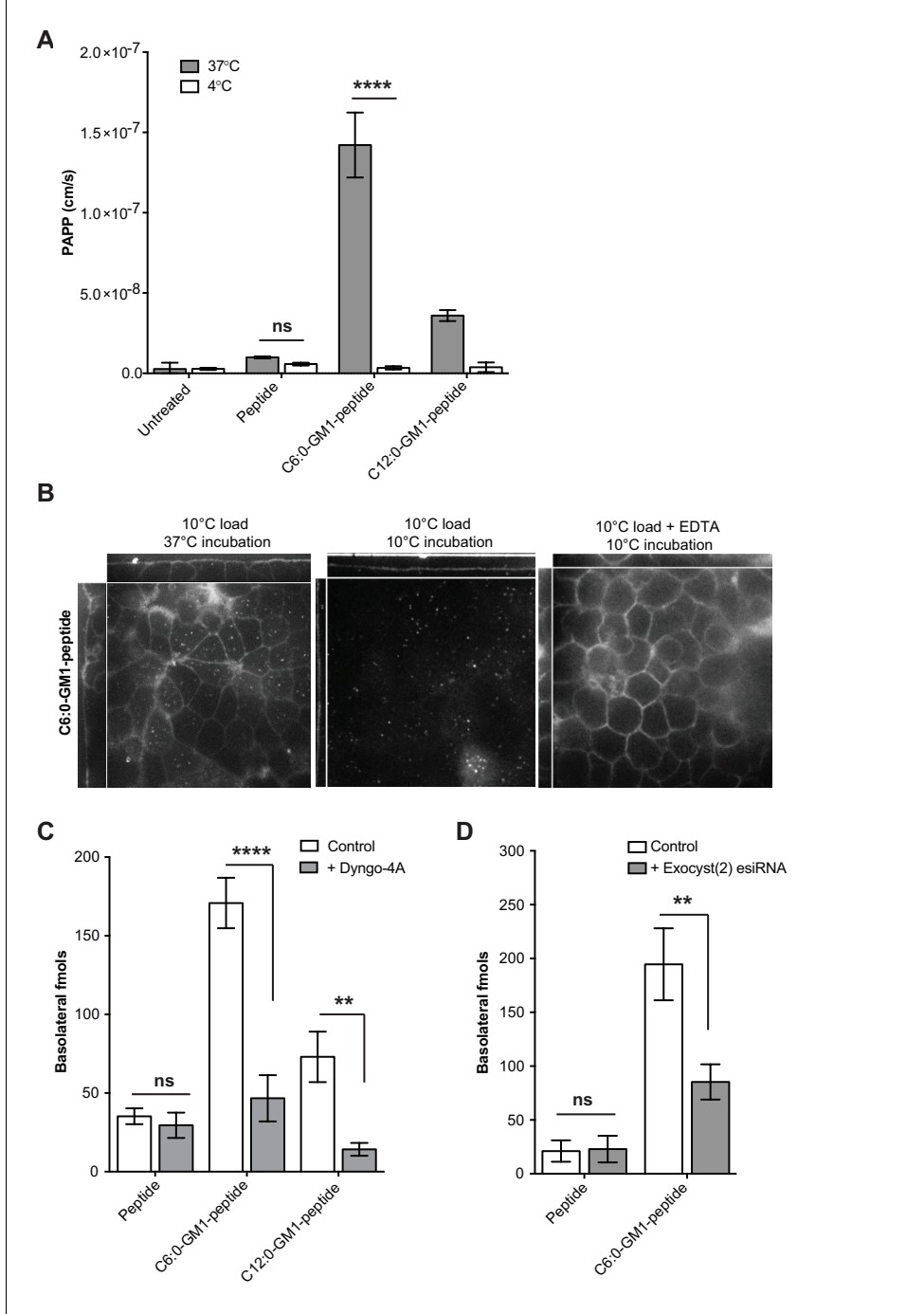

**Figure 2.** Transport of short-chain GM1 ceramides occurs via transcytosis. (**A**) Transport of C6:0 and C12:0 GM1-peptide fusions across T84 cell monolayers reported as apparent permeability (PAPP) shows that minimal transport occurs during a 4°C temperature block. (**B**) Analysis of MDCK-II monolayers loaded with 0.5 μM C6:0-GM1 peptide fusion in the absence (left and middle panels) or presence (right panel) of 2 mM EDTA. Live cell confocal imaging shows minimal transport occurs during a 10°C temperature block (middle panel) consistent with transcellular transport by membrane trafficking and minimal paracellular leak. In the presence of disrupted tight junctions (i.e. in the presence of EDTA, right panel) basolateral membranes are stained by paracellular passive diffusion (transcellular leak). (**C**) Transcytosis across MDCK-II monolayers is blocked by dynamin inhibition of endocytosis (50 μM Dyngo-4A). In Dyngo-4A treated cells (gray bars), there is a significant decrease in transepithelial transport of both C6:0 (n = 8) and C12:0 (n = 8) GM1-peptide fusions but not the unfused reporter peptide (n = 6). (**D**) Transcellular transport of the C6:0-GM1 peptide fusion (n = 6) or unfused reporter peptide (n = 5) in cell

*Figure 2 continued on next page*

*Figure 2 continued*

monolayers depleted of the exocyst complex by esiRNA transfection against EXO2 (gray bars). (mean ± s.e.m)
(ns = non significant, **p<0.01, ****p<0.0001; Bonferroni's multiple comparison test).
DOI: https://doi.org/10.7554/eLife.34469.006

reservoir was strongly inhibited by Dyngo-4A treatment, consistent with active transcellular transport by transcytosis (*Figure 2C*). In contrast, Dyngo-4A had no detectable effect on transport of the reporter peptide alone, as expected for diffusion of solutes by paracellular leak. Similar results were obtained using a genetic approach. The exocyst complex is necessary for efficient receptor-mediated transcytosis of immunoglobulins (*Oztan et al., 2007*; *Nelms et al., 2017*), and esiRNA knockdown of EXOC2 subunit caused the predicted 50% decrease in trans-epithelial transport of the C6:0-GM1 peptide fusion molecule (*Figure 2D*). In contrast, transport of the unfused reporter peptide was not affected by exocyst KD. Thus, fusion of a peptide cargo to certain GM1 species enables active transport of the peptide across the epithelial barrier by transcytosis.

## Amplified rates of release into solution by the very short chain GM1 species

To explain how the very short chain fatty acids amplified transport across epithelial cell monolayers, we first measured the rate of transcytosis by pulse chase. Apical membranes of MDCK cell monolayers were loaded at 10°C with equal amounts of C12:0 and C6:0-GM1-peptide fusions, washed, and then incubated at 37°C and imaged by live-cell confocal microscopy over time. Transcytosis was measured as fluorescence at basolateral membranes. By this method, we found no detectable difference among the two GM1 species in the rate of transcytosis (*Figure 3—figure supplement 1A*). In both cases, basolateral membranes were fluorescent after a 10 min chase. At longer chase times, however, we observed a dramatic difference between the C6:0- and C12:0-peptide fusion molecules (*Figure 3A*). Monolayers loaded with the C12:0-GM1 peptide fusion stained brightly at both the apical (bottom left panel) and basolateral membranes (bottom right panel). In stark contrast, monolayers loaded with the C6:0-GM1 peptide fusion showed no fluorescence (middle panels). We interpreted this result as indicating a higher rate of release from cell membranes to the basolateral solution and emptying the cell of the peptide-GM1 fusion over time. To test this idea, we quantified the rate of release from cell membranes for the fluorescent GM1-peptide fusion molecules (Material and Methods). We studied the rate of GM1 release into DMEM media alone (*Figure 3B*), as well as into DMEM containing defatted bovine albumin (BSA) (*Pagano, 1989*) (*Figure 3C*). Results show a faster and more complete diffusion from membrane to solution for the C6:0-GM1 fusion molecule (*Figure 3B and C*; red curve) compared to the longer chain C12:0-GM1-peptide (blue curve). Faster and more complete release into solution was also observed for the C2:0-GM1-peptide fusion molecule (*Figure 3B and C*; purple curve). Thus, the greater efficiency for transepithelial transport by the short chain GM1 species is largely explained by their greater efficiency of diffusion from membrane to solution after transcytosis.

Glycosphingolipids contain another major functional domain in addition to the ceramide, the extracellular oligosaccharide head group. These are structurally diverse and operate in a variety of biologic activities (*Cantù et al., 2011*). In all cases, however, the oligosaccharide head group acts to trap sphingolipids in the outer leaflet of cell membranes, thus rendering the lipids dependent on membrane trafficking for their distribution across the cell. To test if the effects of ceramide structure on transcytosis and membrane-release were specific to GM1 glycosphingolipids, or could be generalized to other glycosphingolipid species, we fused our reporter peptide to a GM3 ganglioside synthesized to contain ceramide domains with either C12:0 or C6:0 fatty acids. The oligosaccharide domain of GM3 differs from GM1 by the absence of two sugars and thus lacks the terminal galactose (and GalNAc) that functions strongly as a lectin-binding site in GM1. When tested for transcytosis, we unexpectedly found that the GM3-C12:0-peptide fusion molecule crosses epithelial monolayers far more efficiently that the closely related GM1-C12:0-peptide; and as efficiently as the GM1-C6:0 and C2:0 species (*Figure 3D*). Similarly, transepithelial transport for the GM3-C6:0-peptide was approximately 2-fold greater than that observed for the GM1-C6:0-peptide when compared directly (*Figure 3—figure supplement 1B*). Transport was strongly inhibited by pretreatment

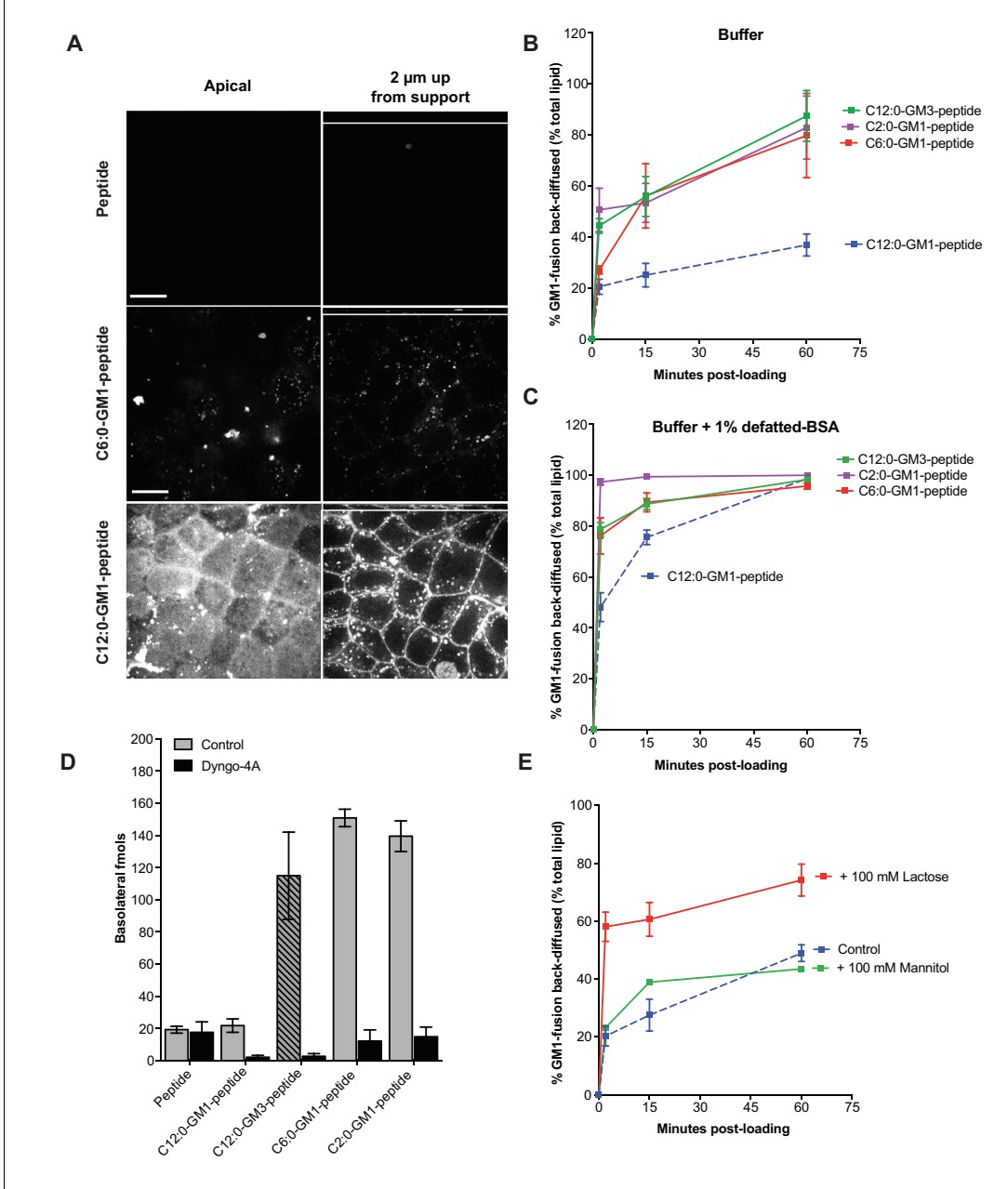

**Figure 3.** Release from membranes to solution by short-chain GM1 species. (A) Live cell confocal images of MDCK-II cell monolayers after a 10 min pulse and 3 hr chase with 0.5 µM reporter peptide, C6:0 or C12:0-GM1 peptide fusion. Images are taken at the plane of apical membranes (left column) or midway through cell body (right column) (B-C) Time courses of diffusion from MDCK-II cell membranes to media (B) or media containing defatted-BSA (C) of C2:0 (n = 5), C6:0 (n = 8), C12:0 (n = 12) GM1-peptide fusions and C12:0-GM3-pepide fusion (n = 9). (D) The C12:0-GM3 peptide fusion has enhanced transepithelial transport across T84 cell monolayers compared to the C12:0-GM1-peptide fusion and is dynamin dependent (n = 4). (E) The rate of membrane release of the C12:0- GM1-peptide molecule is enhanced in the presence of 100 mM lactose, but not 100 mM mannitol implicating a galactose-specific lectin membrane anchor (n = 10) (mean ± s.e.m).

DOI: https://doi.org/10.7554/eLife.34469.007

The following figure supplement is available for figure 3:

**Figure supplement 1.** Membrane release to solution depends on ceramide and oligosaccharide structure.
DOI: https://doi.org/10.7554/eLife.34469.008

with the dynamin inhibitor Dyngo-4A, implicating active transcellular trafficking by transcytosis. In membrane-release studies, we found a higher rate of release to solution for the C12:0-GM3-peptide fusion (green curve) when compared to the GM1 fusion molecule (*Figure 3B and C*). Thus, the GM1 glycosphingolipid species appear to be retained in the membrane more tightly than the GM3 species containing the same ceramide domains. Because GM3 lacks a free terminal galactose, we hypothesized the GM1 lipids, which contain the terminal galactose, might be further tethered to the membrane by a form of lectin-binding at the cell surface. To test this idea, we studied the rate of membrane release for the C12:0-GM1 species in the presence or absence of 100 mM lactose (Glc-Gal disaccharide) as a competitive ligand (*Figure 3E*). These studies show enhanced release from the membrane in the presence of excess free lactose, but not excess mannitol, implicating interaction with a galactose-specific lectin membrane tether (*Figure 3E*). To confirm this idea, we studied the rate of membrane release for the C12:0-GM3 species that lacks the terminal disaccharide galactosyl N-acetyl galactosamine (Gal-GalNAc) contained in GM1 linked (1→4) to the central galactose. Here we find that lactose at high 100 mM concentrations competed both the GM1 and GM3 species off the membrane (not shown) but at lower doses (5 mM) lactose released only the GM1-fusion (*Figure 3—figure supplement 1C–D*). Likewise, Gal-GalNAc (5 mM) was effective at enhancing release of only the GM1-fusion molecules (*Figure 3—figure supplement 1E–F*). Thus, the oligosaccharide domain of the glycosphingolipids can also affect the efficiency of transport across epithelial barriers, we propose by interacting with lectin-like molecules at the cell surface. The results also strengthen the implications and general principles for how the oligosaccharide head group of native glycosphingolipids may affect sorting and retention in specific regions of the cell.

## Glycosphingolipid-mediated absorption across epithelial barriers in vivo

To test for glycosphingolipid-mediated transport across the intestine in vivo, the unfused reporter peptide or the C4:0-GM1-peptide fusion molecule were intragastrically gavaged to mice at equal doses (0.5 nmol/kg) and absorption into the blood analyzed after 15 and 30 min using the streptavidin-capture assay. At both time points, we find evidence of absorption into the systemic circulation for the GM1-peptide fusion molecules (nearly 3% of the applied dose), but not for the unfused peptide (*Figure 4A*). The same results were obtained for the C12:0-GM3-peptide fusion molecule (*Figure 4B*). We also measured uptake into the liver, where at 1 hr after gastric gavage we find the glycosphingolipid-peptide fusion molecule, but not the unfused peptide (*Figure 4C*). Thus, fusion to the glycosphingolipids facilitated absorption of the peptide cargo across the intestine and into the two tissues we sampled, blood and liver. The reporter peptide on its own was not detectably absorbed.

To test if these results can be generalized, we applied the C6:0- and C12:0-GM1-peptide fusions to the nasal epithelium, another tight epithelial mucosal surface. In this case, the C6:0-GM1-peptide (green labeling; *Figure 4D*) could be visualized by two-photon microscopy within the epithelial barrier in all regions of the nasal epithelium (*Figure 4D*), including in areas of pseudostratified (top right panels) and simple columnar epithelial tissues (bottom right panels). Uptake of the unfused peptide, applied at the same dose, was very rarely detected (left panels). Absorption to the systemic circulation for the GM1-peptide fusion molecules was confirmed biochemically by measuring content in the blood 15 min after nasal administration (*Figure 4E*). Here, we find approximately a 10-fold increase in blood levels for the GM1-peptide fusion molecules, compared to peptide alone (which is close to background). Unexpectedly, in the nasal epithelium, we find evidence for efficient absorption of the C12:0-GM1-peptide fusion molecules, similar to our results with the C12:0-GM3-peptide species in the intestine. The result suggests that different tissues may interact in different ways with the oligosaccharide domains of glycosphingolipids. In this case, the nasal epithelium may not bind the GM1 oligosaccharide, thus allowing for more efficient release from cell membranes into solution after transcytosis and systemic absorption.

## Application of the C6:0-GM1 species to enable oral absorption of the incretin hormone GLP-1

Glucagon-like peptide-1 (GLP-1) and related peptides have become important drugs in the management of type two diabetes mellitus, by both promoting weight reduction and sensitizing glucose-stimulated insulin release (*van Bloemendaal et al., 2014*; *Heppner and Perez-Tilve, 2015*;

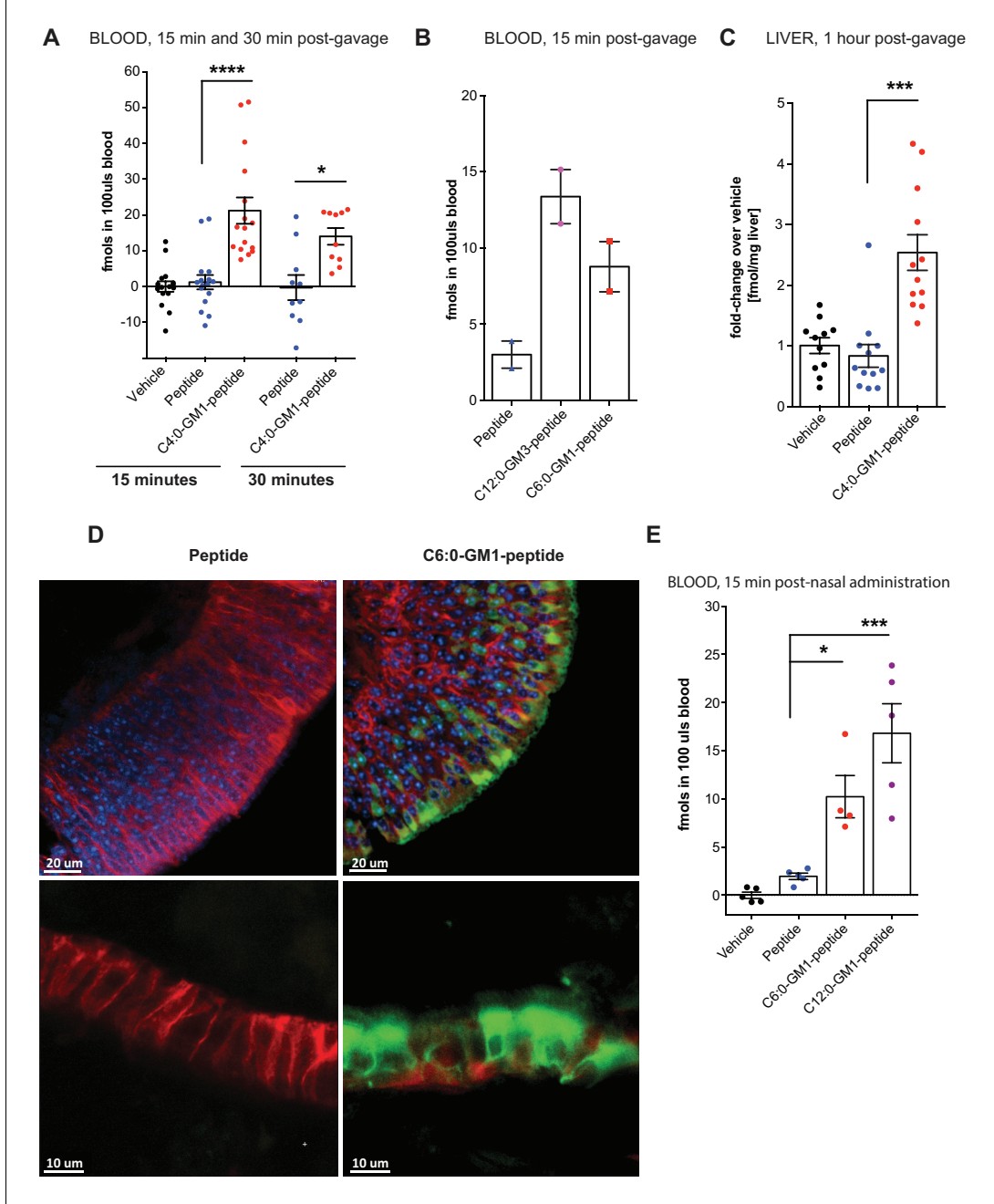

**Figure 4.** Absorption across intestinal and nasal epithelial barriers in vivo. (**A**) In vivo studies showing absorption across intestinal epithelial barriers into blood after gastric administration of the C4:0-GM1-peptide fusion, vehicle alone, or unfused reporter peptide (five independent experiments). (**B**) Absorption across the intestine into blood 15 min after gastric administration of the C12:0-GM3, C6:0-GM1 peptide fusions or unfused reporter peptide (n = 2). (**C**) 1 hr after gastric gavage the C4:0-GM1 peptide fusion is absorbed to the liver whereas the unfused reporter peptide is not detected (four independent experiments). (**D**) Uptake into nasal epithelium. After topical nasal administration, tissue was fixed with 4% formaldehyde and stained with anti-EpCAM to label epithelium (red) and DAPI (blue) for nuclei. Images by two-photon microscopy comparing transport of the unfused reporter peptide (left panels) and C6:0-GM1 peptide fusion (green labeling, right panels) Scale bars 20 um upper panels, 10 um lower panels. (**E**) Biochemical analysis of blood 30 min after nasal administration shows systemic absorption of C6:0 and C12:0-GM1 peptide fusions (two independent experiments). (**A–E**) Each data point on graphs represents individual mice and bars represent mean ± s.e.m. (ns = non significant, *p<0.5; ***p<0.0001, Tukey's multiple comparison test).

*Figure 4 continued on next page*

*Figure 4 continued*

DOI: https://doi.org/10.7554/eLife.34469.009

The following source data is available for figure 4:

**Source data 1.** Summary Data for *Figure 4A*.

DOI: https://doi.org/10.7554/eLife.34469.010

**Source data 2.** Summary Data for *Figure 4E*.

DOI: https://doi.org/10.7554/eLife.34469.011

*Tran et al., 2017*). A major factor limiting the clinical utility in many individuals is the fact that all currently available preparations must be delivered by subcutaneous injection. To test if the properties of glycosphingolipid trafficking could be applied to enable oral absorption of GLP-1, we coupled a long-half-life version of GLP-1 (*Figure 5A*) with C-terminal peptide linker (termed here GLP-1 for simplicity) to the C6:0-GM1 ceramide species as described (*te Welscher et al., 2014*). The bioactivity of the glycolipid-GLP-1 fusion molecule was quantitatively assessed using HEK cells expressing the hGLP-1 receptor and CRE (cAMP) luciferase reporter (*te Welscher et al., 2014*). As controls, the commercially available long-acting GLP-1 (Exendin-4), and the unfused GLP-1-peptide were assessed in parallel (*Figure 5—figure supplement 1A*). The fusion of C6:0-GM1 to GLP-1 caused some loss of function, but the molecule remained highly potent as an incretin hormone, closely comparable to that of the controls.

The GLP-1 peptide cargo is 40 residues, approximately 4-fold greater in size compared to the reporter peptide. We first studied transport across intestinal T84 cell monolayers in vitro to test if GM1 glycosphingolipids could transport such a larger cargo. In these studies, GLP-1 transport was quantified by luciferase bioassay as previously described (*te Welscher et al., 2014*) (*Figure 5B*). Here, we find an even greater effect of the glycosphingolipids on transepithelial GLP-1 transport (20–100-fold above controls). This is explained by a much lower rate of paracellular leak for the larger sized 40-residue GLP-1 peptide. Such size-exclusion from tight junctions is a well-known determinant of paracellular solute diffusion across intact epithelial barriers.

To test for absorption and biologic incretin activity in vivo, we gastrically gavaged equal doses (10 nmol/kg) of the C6:0-GM1-GLP-1 fusion, the unfused GLP-1 peptide (GLP-1 oral), or vehicle into wild-type mice and measured effects on glucose metabolism by glucose tolerance test (*Figure 5C*). Here, we find a lower peak and more rapid return of blood glucose to normal levels in the animals gavaged the C6:0-GM1-GLP-1 fusion molecules compared to animals gavaged GLP-1 peptide (*Figure 5C and D*). The effect on glucose tolerance by gastrically administered C6:0-GM1-GLP-1 was similar to the effect achieved by the intraperitoneal injection of GLP-1 peptide alone, implicating an equally high level of bioavailability for the gastrically-delivered GM1-fusion molecule (*Figure 5D*).

We confirmed intestinal absorption of the C6:0-GM1-GLP-1 into the systemic circulation in two ways. First, we measured GLP-1 activity in blood samples by streptavidin capture and quantitative luciferase bioassay (*Figure 5E*). The results show absorption of the GM1-GLP-1 fusion molecule into the blood, but not for unfused GLP-1. In a second approach, we synthesized an all D-amino acid (non-degradable) isomer of GLP-1 coupled to AF488 to allow for direct quantitative measurement of the 40-residue isomer in the blood using the same streptavidin capture assay as described for our reporter peptide (*Figure 5A*). Again, we find evidence for absorption of the GLP-1 cargo when fused to the C6:0-GM1 transport vehicle, but no absorption for the unfused GLP-1 peptide (*Figure 5F*). These experiments were performed in two different laboratories, using two different animal facilities with the same results. In all assays (*Figure 5C–5F*), we find the efficiency of intestinal absorption enabled by fusion to C6:0-GM1 was again almost as efficient as for IP injection of the peptide alone, implicating a high level of bioavailability for the GM1-fusion molecules applied by gastric gavage.

Notably, however, the C2:0-GM1-GLP-1 fusion molecule had no effect on glucose tolerance (*Figure 5D*) and was not detectably absorbed after gastric gavage (*Figure 5—figure supplement 1B*), even though this molecule was readily transported across epithelial monolayers in vitro (*Figure 1C*). This may be explained by lower affinity of the C2:0- (and lyso-) ceramide domains for incorporation into cell membranes, as inferred from membrane loading and release assays (*Figure 3B* and *Figure 1—figure supplement 2D*). The difference in biology (transcytosis in vitro versus absorption in vivo) becomes apparent only in vivo where the conditions for epithelial uptake

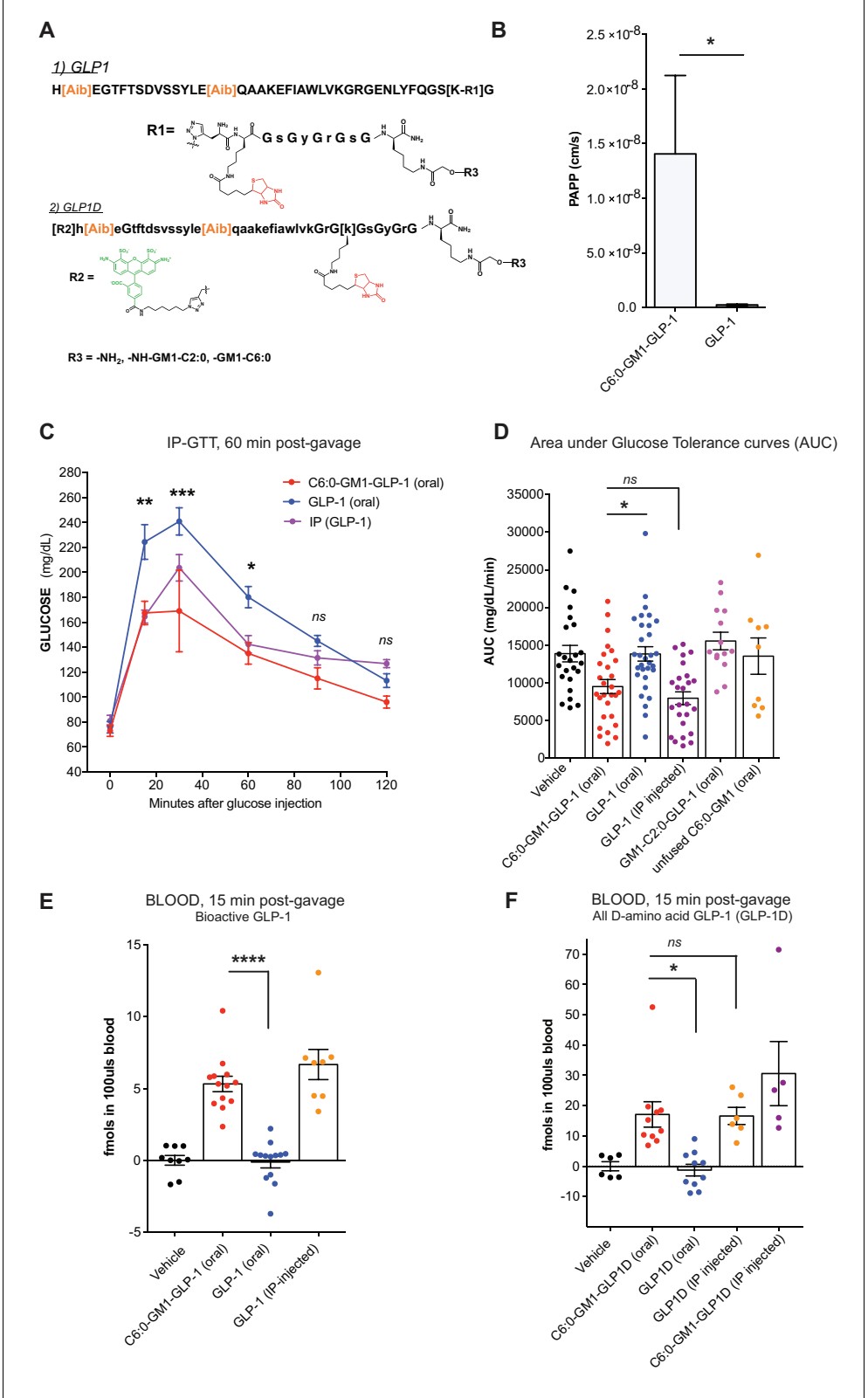

**Figure 5.** GM1-mediated absorption of GLP-1 affects blood glucose metabolism. (**A**) GLP-1 and all-D GLP-1 isomer sequence used for coupling to a C6:0 and C2:0 GM1 ceramide species. (**B**) In vitro transcytosis assay with C6:0-GM1-GLP-1, or unfused GLP-1 across T84 cell monolayers (three independent experiments) (Unpaired t-test, *p<0.5). (**C**) Representative intraperitoneal glucose tolerance test after gastric gavage of 10 nmol/kg C6:0-GM1-

*Figure 5 continued*

GLP-1. Each point represents mean ± s.e.m (n = 4 mice). Mice fed C6:0-GM1-GLP-1 (red curve) show faster recovery after a glucose challenge in contrast to mice gavaged with unfused GLP-1 (blue curve) or vehicle (black curve) (**D**) Effect of the indicated GM1-GLP-1 species, GLP-1 alone, or vehicle on glucose tolerance quantified as Area under Curve (AUC) for 10 independent experiments with each data point representing individual mice and bar representing the mean ± s.e.m. (**E**) GLP-1 in blood 15 min after gastric gavage quantified for each species using the luciferase bioassay (fmols of compound per 100uls blood for three independent experiments). (**F**) Systemic absorption of an all D-isomer of GLP-1 used to directly measure the cargo in blood (three independent experiments). (**A–E**) Each data point on graphs represents individual mice and bars represent mean ± s.e.m. (ns = non significant, *p<0.5; ***p<0.0001, Tukey's multiple comparison test).

DOI: https://doi.org/10.7554/eLife.34469.012

The following source data and figure supplement are available for figure 5:

**Source data 1.** Source Data for *Figure 5D*.
DOI: https://doi.org/10.7554/eLife.34469.014
**Source data 2.** Source Data for *Figure 5E*.
DOI: https://doi.org/10.7554/eLife.34469.015
**Source data 3.** Source Data for *Figure 5F*.
DOI: https://doi.org/10.7554/eLife.34469.016
**Figure supplement 1.** GM1-mediated absorption of GLP-1.
DOI: https://doi.org/10.7554/eLife.34469.013

and transport are not optimized as they are in vitro. Thus, although it seemed at first glance that further shortening of the fatty acid beyond C4:0 should amplify transepithelial transport and thus clinical utility, this was not the case and the result informs further development of the technology.

In summary, we find that fusion of therapeutic peptides to GM1 and GM3 glycosphingolipids with short fatty acids enables their active transport across tight epithelial barriers by transcytosis. In the case of the incretin hormone GLP-1, fusion to the lipid carriers allows for gastric (oral) absorption with high bioavailability and the expected effects on blood glucose, highlighting the potential use of this technology in clinical applications.

## Discussion

Our findings delineate a novel synthetic method for enabling absorption of therapeutic peptides across mucosal surfaces in vivo. The approach is based on the natural biology of lipid sorting for the glycosphingolipids, which depends primarily on the structure of the ceramide domain to allow for trafficking in the transcytotic pathway, and thus active transport across mucosal surfaces without barrier disruption. For applications requiring systemic drug delivery, non-native glycosphingolipid carriers with ceramide domains containing short-chain fatty acids are required to allow for efficient release from cell membranes into the circulation after transcytosis. The apparent high level of intestinal bioavailability enabled by the glycosphingolipid carriers is unprecedented.

The mechanism(s) for transcellular trafficking co-opted by the *cis*-unsaturated or short chain fatty acid glycosphingolipids are not fully understood. The most robust sorting event for GM1 glycosphingolipids appears to occur in the early endosome where long saturated chain ceramides are trafficked to the late endosome/lysosome, and the *cis*-unsaturated and short-chain glycoceramides are not ([*Chinnapen et al., 2012*] and Schmieder and Lencer unpublished results). It is possible the unsaturated and short-chain ceramide domains engage sorting mechanisms that dictate their trafficking to the recycling endosome and elsewhere, but we suggest it also possible that their trafficking might be stochastic after escape from the lysosomal pathway, essentially tracking along with bulk membrane flow. In other words, the robust sorting event may occur only for the long chain saturated glycosphingolipids, directing them to the lysosome.

Another key structural feature enabling this technology must be the oligosaccharide head group. This domain traps the ceramide lipid in the outer membrane leaflet, preventing flip-flop between leaflets and thus rendering the molecule dependent on membrane dynamics for movement throughout the cell – an essential feature for a trafficking vehicle. As shown by our studies using GM3, the extracellular oligosaccharide can in some cell types also affect the efficiency of transepithelial

transport. One way, we suggest, may be by binding to adjacent membrane lectins, thus enhancing the tethering of the lipid to the membrane surface. Differences have also been reported between GM1 and GM3 with respect to plasma membrane localization and bilayer/curvature dynamics in vitro (*Cantù et al., 2011*; *Janich and Corbeil, 2007*).

In the case of transport across mucosal barriers, we envision several applications for the glycosphingolipids of relevance to clinical medicine. One would be as vehicles for systemic delivery of peptide hormones as demonstrated here, or for topical delivery of agonist or antagonist peptides to specific mucosal surfaces. Another would be for delivery of antigens or adjuvants to enable mucosal vaccination or oral tolerance. It is possible that these glycosphingolipids will transport therapeutic proteins in the same way. Finally, while the biology of endosomes in endothelial cells is much less well understood, we believe at least some of the basic principles for lipid sorting in epithelial cells will apply to this cell type; and the glycosphingolipid carriers defined here may also be used to enable transport of biologics across tight endothelial barriers.

## Materials and methods

### Cell lines

T84 cells and MDCK cells were both obtained from ATCC (American Type Culture Collection). T84 cells were cultured in a 1:1 mixture of Ham's F12 medium and Dulbecco's modified Eagle's medium with 2.5 mM L-glutamine, 95%; fetal bovine serum, 5%. MDCK cells were maintained in DMEM supplemented with 10% heat-inactivated FBS and penicillin and streptomycin, all obtained from Thermo Fisher Scientific. T84 and MDCK cells were routinely confirmed to be negative for mycoplasma using a PCR kit from MD Biosciences.

### Transcytosis assay

T84 or MDCK-II cells plated on 24-well Transwell inserts (polyester membranes, Costar) were washed and equilibrated in DMEM without serum containing defatted-BSA (df-BSA). Unconjugated reporter peptide or GM1-peptide fusions at 0.1 μM complexed to df-BSA in a 1:1 ratio were then added apically for 3 hr. An excess of BSA (1% wt volume) was added basolaterally to aid in extraction of lipid from cell membranes. After a 3 hr continuous incubation, 1 mL basolateral media was collected and incubated with 10 μL magnetic streptavidin sepharose beads overnight at 4°C, washed with TBS-Tween, and eluted in 95% formamide/10 mM EDTA/0.4 mg/mL biotin. For detection of the reporter peptide or GM1-peptide fusion, fluorescence was read using an Infinite M1000 plate reader (Tecan). For each biological replicate concentrations were calculated from standard curves for each compound.

The apparent permeability coefficient (PAPP, cm/s) is calculated across cell monolayers grown in transwells based on the appearance rate of lipid-peptide fusions in the basolateral compartment over time:

Papp (cm/s) = (VD / (A*MD))* (DMR/Dt)

Papp (cm/s)= ($cm^3$ / ($cm^2$ * mol)) * (mol/s)

VD = apical (donor) volume ($cm^3$)

MD = apical (donor) amount (mol)

A = membrane surface area ($cm^2$) of apical (donor) chamber (i.e. transwell surface area)

DMR/Dt = the amount of compound (mol) transferred to the basolateral (receiver) compartment over time (s).

The reader is referred to Bio-Protocol (*Garcia-Castillo et al., 2018*).

### In vivo studies

WT C57/BL/6 mice (male 7–9 weeks old) were purchased from Jackson Laboratory (Maine USA) and acclimatized for one week. For intestinal absorption experiments, mice that were fasted overnight were lightly anesthetized with isoflurane and gastrically gavaged with a 0.5 nmol/kg dose in a 200 μL volume. Compounds were diluted in PBS containing df-BSA in a 1:1 ratio prior to administration to mice. For analysis of systemic absorption, blood samples were obtained using standard cardiac puncture procedures at 15 or 30 min after compound administration. 100 μL blood was diluted with

400 µL RIPA buffer and incubated with 10 µL streptavidin Sepharose overnight at 4°C, washed, and eluted in 95% formamide/10 mM EDTA/0.4 mg/mL biotin as in our in vitro assay.

Liver tissue was flash frozen in liquid nitrogen and ground with a chilled mortar and pestle on dry ice. After obtaining dry weight, samples were homogenized in 1 mL RIPA buffer, centrifuged, and supernatant incubated with 10 µL streptavidin Sepharose and bound molecules eluted with 95% formamide/10 mM EDTA/0.4 mg/mL biotin. Amount of compound accumulated in the liver was normalized per mg dry weight.

For intraperitoneal glucose tolerance tests, a 10 nmol/kg dose was used to gavage overnight-fasted WT C57/BL/6 mice (male 7–9 weeks old) with GM1-GLP-1 fusion molecules or unfused GLP-1. Glucose measurements following i.p administration of 2 mg/g glucose solution were obtained from tail vein blood applied directly to glucose strips as in (te Welscher et al., 2014).

## Membrane loading and lipid release into solution

MDCK-II cells were plated on 96-well plates the day prior to the experiment. Cells were washed with 10°C serum-free DMEM (no phenol red) and equilibrated with DMEM containing 0.1 µM df-BSA for 15 min. Cells were loaded for 45 min at 10°C with 0.1 µM GM1-peptide molecule with a molar ratio of 1:1 (lipid:df-BSA). After loading, cells were washed, warmed to 37°C degrees in DMEM (no phenol red) to allow for proper lipid incorporation, and incubated with 0.25% trypsin in HBSS to release adherent glycosphingolipids not incorporated into the membrane bilayer.

Cells were then incubated in DMEM alone or DMEM containing 1% df-BSA for 2 min, 15 min, or 1 hr. After the indicated time, media was collected and GM1-peptide molecules released into solution quantified using standards for each compound. Cells were subsequently lysed in RIPA buffer and the amount of cell-associated GM1-peptide remaining at that time point quantified using known standards. Amount of GM1-peptide released into the solution was calculated as a ratio of total lipid incorporated (i.e GM1-peptide in media + cell associated GM1-peptide).

## Synthesis of Ganglioside-Peptide conjugates

Gangliosides of different fatty acid species were supplied by Prof. Sandro Sonnino (U. Milan, Italy). Peptides containing modified functional residues were custom synthesized by Novo Nordisk (DK). Synthesis of peptide-lipid conjugates was accomplished by a modified method previously published (te Welscher et al., 2014). In a typical 2 mL reaction, 2 mg (approximately 1300 nmoles depending on fatty acid) of ganglioside was oxidized with sodium periodate (13 µmoles) in oxidation buffer (100 mM sodium acetate pH 5.5, 150 mM NaCl) for 30 min on ice and protected from light. The reaction was quenched by addition of glycerol (5% final). The reaction was desalted by Bond Elut SepPak C18 cartridge (Agilent, MA) and methanol used to elute from the column was removed by Speed Vac concentration (Savant). The oxidized product was then reconstituted in 2 mL PBS pH 6.9 in the presence of 10% DMF and reacted with 2700 nmoles of aminooxy-containing peptide in the presence of 10 mM aniline (Dirksen and Dawson, 2008). The reaction was incubated for 20 hr at room temperature with mixing on a nutator, where the GM1-peptide fusion product formed normally resulted in a white precipitate. The precipitate was separated from the solution by centrifugation, then resuspended in 400 µL 50% isopropanol/water after brief sonication. PBS pH 6.9 was added (200 µL) along with 4.8 µmoles of sodium cyanoborohydride and incubated for 3 hr to reduce the oxime bond. Lipid-peptide conjugates were purified by semi-preparative HPLC, and confirmed by either MALDI-TOF (AB Voyager), or ESI LC-MS (Agilent, MA).

## Fluorescent reporter peptides

With exception of the fluorescent peptide described in Extended Data Figure 1 that was done by maleimide linkage, the labeling of peptides with Alexa fluorophore was typically done via copper-mediated Click chemistry. 320 µM peptide-lipid fusions containing an N-terminal alkyne residue (propargylglycine) were reacted with equimolar concentrations of Alexa Fluor 488 -azide under the following conditions. 50 mM Tris-Cl, 5 mM copper (II) sulfate, 100 mM sodium ascorbate, 37 mM (Tris[(1-benzyl-1H-1,2,3-triazol-4-yl)methyl]amine, TBTA in DMSO/t-butanol 1:4) 1 mM (Tris(2-carboxyethyl) phosphine hydrochloride, TCEP – Sigma) and reacted for 16 hr at room temperature with mixing via nutator. Products were purified by HPLC and confirmed by mass spectrometry. Products were lyophilized and stored at −20°C. Compounds were resuspended in 33% DMF/water to make

stock solutions for assays. m/z mass spectrometry values ± 3 Da were as follows: For GM1-C12:0 reporter conjugates with different functional groups on the peptide and d18:1 long chain base, alkyne = 2475.5 Da(1+); biotin = 2734.4 Da(1+); alkyne-biotin = 2829.4 (Da) (1+); Alexa Fluor 488 maleimide = 1552.2 Da and d20:1 = 1566.2 Da. For GM1-C16:0 species in this series, (2+) mass was observed at 1580.2 Da.

Most of the structure-function studies with GM1 fatty acid species lyso to C12:1, were detected with a 3 + charge. m/z values ± 5 Da for d18:1 and d20:1 sphingosine, respectively, were as follows: lyso = 1101.1 Da (3+) and 1110.5 Da (3+); C2:0 = 1115.1 Da (3+) and 1124.5 Da (3+); C4:0 = 1124.5 Da (3+) and 1133.8 Da (3+); C6:0 = 1134.1 Da (3+) and 1142.5 Da (3+); C6:1 = 1133.1 Da (3+) and 1143.5 Da (3+); C12:0 = 1161.8 Da (3+) and 1171.2 Da (3+); C12:1 = 11161.1 Da (3+) and 1170.5 Da (3+). Free peptide was observed as a single ion peak at 2102.8 Da. For GM3 molecular species conjugates, m/z was observed at: C6:0 = 1212.1 Da (3+) and 1026.1 Da (3+); C12:0 = 1040.1 Da (3+) and 1054.1 Da (3+).

## Synthesis of GLP1-ganglioside AND GLP-D conjugates

To generate bioactive GLP1 fusion lipids, two peptides were joined together via a triazole linkage. Long half-life GLP1 sequences were synthesized containing isobutyrate residues substituted at key dipeptidyl peptidase-4 (DPP-4) cleavage sites, and a C-terminal azido-lysine (*Figure 5* and Extended Data *Figure 1*). A Tobacco Etch Virus protease site (ENLYFQS) was originally designed into the sequence but was not used for the purposes of this paper. The peptide was joined to reporter peptide-lipid conjugates via N-terminal alkyne using Click chemistry as described above.

To synthesize the all-D GLP1-lipid fusions, peptides were made as a complete chain on solid phase, and contained aminooxy and biotin groups (*Figure 5*). Linkage to oxidized ganglioside was performed as stated above. m/z values ± 8 Da for the biologic GLP1-fusion were observed at: C2:0 = 1808.4 Da (4+); C6:0 = 1822.2 with free peptide seen as a 3 + charge at 2251.0 Da. For the all-D isomer version of GLP1, GLPD fused to GM1-C6:0, m/z was seen as a 3 + charge at 2251.0 Da and the free peptide as a 2 + charge at 2726.7 Da.

## Acknowledgements

This project was supported by a NIH F-32 DK111072-01 to MDG-C; DK084424, DK048106, DK104868, and an unrestricted innovator grant from Novo Nordisk to WIL; DK104868 to DC.; DK031036 to C RK.; HD000850 to SS.; T32 DK007477 to RJG; and PO1 AI112521 and RO1 AR068383 to Uv-A. We acknowledge the Harvard Digestive Disease Center DK034854 and the Harvard Medical School Center for Immune Imaging. We are grateful to Dr. Elisha Fielding for helping to develop animal tissue harvesting protocols and Kiara Blue for chemical synthesis of lipid-peptide fusions, as well as members of the Lencer laboratory for their helpful discussions. An innovator grant and collaboration with Novo Nordisk was awarded to WIL with no restrictions for initial studies on peptide reporter design (2012–2014). Novo Nordisk declined any and all claims to the technology in 2015.

## Additional information

### Competing interests

Jesper Lau: Jesper Lau is affiliated with Novo Nordisk. The author owns shares in Novo Nordisk but has no financial interests to declare relevant to this manuscript. Wayne I Lencer: An innovator grant and collaboration with Novo Nordisk was awarded to WIL with no restrictions for initial studies on peptide reporter design (2012-2014). Novo Nordisk declined any and all claims to the technology in 2015. The other authors declare that no competing interests exist.

## Funding

| Funder | Grant reference number | Author |
| --- | --- | --- |
| National Institute of Diabetes and Digestive and Kidney Diseases | DK111072-01 | Maria Daniela Garcia-Castillo |
| National Institute of Diabetes and Digestive and Kidney Diseases | DK104868 | Daniel J.-F. Chinnapen<br>Wayne I Lencer |
| National Institute of Diabetes and Digestive and Kidney Diseases | DK048106 | Wayne I Lencer |
| National Institute of Diabetes and Digestive and Kidney Diseases | DK31036 | C. Ronald Kahn |
| National Institute of Diabetes and Digestive and Kidney Diseases | HD000850 | Samir Softic |
| National Institute of Diabetes and Digestive and Kidney Diseases | DK007477 | Rodrigo J Gonzalez |
| National Institute of Allergy and Infectious Diseases | AI112521 | Ulrich von Andrian |
| National Institute of Arthritis and Musculoskeletal and Skin Diseases | AR068383 | Ulrich von Andrian |
| National Institute of Diabetes and Digestive and Kidney Diseases | DK034854 | Wayne I Lencer |

The funders had no role in study design, data collection and interpretation, or the decision to submit the work for publication.

## Author contributions

Maria Daniela Garcia-Castillo, Daniel J-F Chinnapen, Ulrich H von Andrian, Wayne I Lencer, Conceptualization, Resources, Data curation, Formal analysis, Supervision, Funding acquisition, Validation, Investigation, Visualization, Methodology, Writing—original draft, Project administration, Writing—review and editing; Yvonne M te Welscher, Data curation, Investigation, Methodology, Writing—review and editing; Rodrigo J Gonzalez, Data curation, Formal analysis, Investigation, Methodology, Writing—review and editing; Samir Softic, Resources, Validation, Investigation, Methodology, Writing—review and editing; Michele Pacheco, Resources, Data curation, Formal analysis, Investigation, Methodology, Writing—review and editing; Randall J Mrsny, Bradley L Pentelute, Conceptualization, Resources, Validation, Investigation, Methodology, Writing—review and editing; C Ronald Kahn, Resources, Data curation, Formal analysis, Validation, Investigation, Methodology, Writing—review and editing; Jesper Lau, Resources, Data curation, Investigation, Methodology, Writing—review and editing

## Author ORCIDs

Maria Daniela Garcia-Castillo http://orcid.org/0000-0002-8456-2531
Daniel J-F Chinnapen http://orcid.org/0000-0003-3100-3346
Wayne I Lencer http://orcid.org/0000-0001-7346-2730

## Ethics

Animal experimentation: This study was performed in strict accordance with the recommendations in the Guide for the Care and Use of Laboratory Animals of the National Institutes of Health. All of the animals were handled according to approved procedures and protocols by the Boston Children's Hospital Institutional Animal Care and Use Committee (IAUC protocol #15-05-2945R).

Decision letter and Author response
Decision letter https://doi.org/10.7554/eLife.34469.019
Author response https://doi.org/10.7554/eLife.34469.020

## Additional files

### Supplementary files

• Transparent reporting form
DOI: https://doi.org/10.7554/eLife.34469.017

### Data availability

All data analysed during this study are included in the manuscript and supporting files. Source data files have been provided for mouse experiments in Figures 4 and 5.

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
