## [Decision Letter]

Thank you for submitting your article "Mucosal absorption of therapeutic peptides by harnessing the endogenous sorting of glycosphingolipids" for consideration by *eLife*. Your article has been favorably evaluated by Vivek Malhotra (Senior Editor) and three reviewers, one of whom, Christopher G Burd (Reviewer #1), is a member of our Board of Reviewing Editors. The following individual involved in review of your submission has agreed to reveal their identity: Ilya Levental (Reviewer #2).

The reviewers have discussed the reviews with one another and the Reviewing Editor has drafted this decision to help you prepare a revised submission.

Summary: Garcia-Castillo have investigated the utility of glycosphingolipid conjugates as vehicles for delivering macromolecules across cultured cell monolayers and murine mucosal epithelia. The foundation of the strategy lies in published work that established fatty acid-based inter-organelle trafficking determinants of ceramides and ceramide conjugates which showed that unsaturated or short chain unsaturated fatty acid moieties confer sorting into recycling and transcytosis pathways. This study demonstrates that the head group of such ceramide molecules can be substantially modified, including by conjugation to a bioactive molecule, and still retain intracellular sorting characteristics, including transcytosis to the basolateral membrane of cultured cells and in a murine model.

Essential revisions:

The reviewers found this to be a superbly conducted, innovative study that establishes a potentially useful therapeutic drug delivery strategy. There is one point (#1) that the reviewers agreed was not substantiated and this will need to be resolved. We note that the point does not challenge the main conclusion of the manuscript.

1) The premise of the lactose experiment shown in Figure 3E is problematic. Both GM1 and GM3 have a lactose core, so any putative lectin affected by lactose should recognize GM1 and GM3 similarly, so the experiment does not directly address the role of a terminal galactose on release from the membrane. The reviewers feel that your conclusion should be tested with an additional competition experiment that examines a GM1-specific sugar (e.g., GalNac), or less directly, neuraminidase treatment to convert GM1 into a different structure, which should therefore affect lectin binding. In addition, it is advisable to include a test the effect of lactose on a GM3-based lipid in Figure 3E.

---

## [Author Response]

Essential revisions:The reviewers found this to be a superbly conducted, innovative study that establishes a potentially useful therapeutic drug delivery strategy. There is one point (#1) that the reviewers agreed was not substantiated and this will need to be resolved. We note that the point does not challenge the main conclusion of the manuscript.1) The premise of the lactose experiment shown in Figure 3E is problematic. Both GM1 and GM3 have a lactose core, so any putative lectin affected by lactose should recognize GM1 and GM3 similarly, so the experiment does not directly address the role of a terminal galactose on release from the membrane. The reviewers feel that your conclusion should be tested with an additional competition experiment that examines a GM1-specific sugar (e.g., GalNac), or less directly, neuraminidase treatment to convert GM1 into a different structure, which should therefore affect lectin binding. In addition, it is advisable to include a test the effect of lactose on a GM3-based lipid in Figure 3E.

These comments were most helpful. Both studies were completed and the results are consistent with our original interpretations.

In the first experiment, we repeated the lactose competition studies, this time comparing the GM1-peptide and GM3-peptide fusion molecules as suggested. We found that at very high doses, lactose (100 mM) enhanced membrane release of both GM1- and GM3-based peptide fusions. At lower doses, however, lactose (5 mM) was specific for the GM1-peptide fusion molecule (Figure 3—figure supplement 1C-D). The result is consistent with the idea that the terminal galactose of GM1 (absent in GM3) may interact with a membrane bound lectin to retain the lipid-peptide fusion in the membrane.

In the second experiment we used the disaccharide galactosyl N-acetyl galactosamine (GalGalNAc) as competing ligand – using the same experimental format. This disaccharide matches the two extra sugars contained in GM1 and absent in GM3. We found the galactosyl N-acetyl galactosamine disaccharide competed the GM1-peptide fusion molecules off the membrane as predicted, but had no effect on the GM3-peptide fusion molecules (Figure 3—figure supplement 1E-F).

These results confirm the original explanation that the terminal galactose (or galactosyl N-acetyl galactosamine dissacharide (Gal-GalNAc)) of GM1 (absent in GM3) may interact with a membrane bound lectin to retain the lipid-peptide fusion in the membrane. They also strengthen the implications and general principles for how the oligosaccharide head group of native glycosphingolipids may affect sorting and retention in specific regions of the cell.

The text has been revised:

“These studies show enhanced release from the membrane in the presence of excess free lactose, but not excess mannitol, implicating interaction with a galactose-specific lectin membrane tether (Figure 3E). […] The results also strengthen the implications and general principles for how the oligosaccharide head group of native glycosphingolipids effects may sorting and retention in specific regions of the cell.”